# Characterizing the Interplay of Rubisco and Nitrogenase Enzymes in Anaerobic-Photoheterotrophically Grown *Rhodopseudomonas palustris* CGA009 through a Genome-Scale Metabolic and Expression Model

Niaz Bahar Chowdhury,[a] Adil Alsiyabi,[a] Rajib Saha[a]

aChemical and Biomolecular Engineering, University of Nebraska-Lincoln, Lincoln, Nebraska, USA

**ABSTRACT** *Rhodopseudomonas palustris* CGA009 is a Gram-negative purple nonsulfur bacterium that grows phototrophically by fixing carbon dioxide and nitrogen or chemotrophically by fixing or catabolizing a wide array of substrates, including lignin breakdown products for its carbon and fixing nitrogen for its nitrogen requirements. It can grow aerobically or anaerobically and can use light, inorganic, and organic compounds for energy production. Due to its ability to convert different carbon sources into useful products during anaerobic growth, this study reconstructed a metabolic and expression (ME) model of *R. palustris* to investigate its anaerobic-photoheterotrophic growth. Unlike metabolic (M) models, ME models include transcription and translation reactions along with macromolecules synthesis and couple these reactions with growth rate. This unique feature of the ME model led to nonlinear growth curve predictions, which matched closely with experimental growth rate data. At the theoretical maximum growth rate, the ME model suggested a diminishing rate of carbon fixation and predicted malate dehydrogenase and glycerol-3 phosphate dehydrogenase as alternate electron sinks. Moreover, the ME model also identified ferredoxin as a key regulator in distributing electrons between major redox balancing pathways. Because ME models include the turnover rate for each metabolic reaction, it was used to successfully capture experimentally observed temperature regulation of different nitrogenases. Overall, these unique features of the ME model demonstrated the influence of nitrogenases and rubiscos on *R. palustris* growth and predicted a key regulator in distributing electrons between major redox balancing pathways, thus establishing a platform for *in silico* investigation of *R. palustris* metabolism from a multiomics perspective.

**IMPORTANCE** In this work, we reconstructed the first ME model for a purple nonsulfur bacterium (PNSB). Using the ME model, different aspects of *R. palustris* metabolism were examined. First, the ME model was used to analyze how reducing power entering the *R. palustris* cell through organic carbon sources gets partitioned into biomass, carbon dioxide fixation, and nitrogen fixation. Furthermore, the ME model predicted electron flux through ferredoxin as a major bottleneck in distributing electrons to nitrogenase enzymes. Next, the ME model characterized different nitrogenase enzymes and successfully recapitulated experimentally observed temperature regulations of those enzymes. Identifying the bottleneck responsible for transferring an electron to nitrogenase enzymes and recapitulating the temperature regulation of different nitrogenase enzymes can have profound implications in metabolic engineering, such as hydrogen production from *R. palustris*. Another interesting application of this ME model can be to take advantage of its redox balancing strategy to gain an understanding of the regulatory mechanism of biodegradable plastic production precursors, such as polyhydroxybutyrate (PHB).

Address correspondence to Rajib Saha, rsaha2@unl.edu.

The authors declare no conflict of interest.

**KEYWORDS** *R. palustris*, ME model, nitrogenase, rubisco, ferredoxin, electron distribution

*R*hodopseudomonas palustris CGA009 (*R. palustris*) is an alphaproteobacterium with the ability to grow in both metabolic niches, phototrophic, and chemotrophic. Furthermore, it can fix both carbon dioxide and nitrogen and can grow under both aerobic and anaerobic conditions by using light and organic or inorganic compounds to generate ATP (1, 2). Using these metabolic versatilities, *R. palustris* has emerged as a potential biotechnological platform for bioremediation (3–5), bioplastics production (6, 7), and bioelectricity generation (8, 9), wastewater treatment (10–12), and hydrogen production (13–17). Furthermore, *R. palustris* is the only known bacteria to encode all three known nitrogenase enzymes (2) besides *Azotobacter vinelandii* (18). *R. palustris* also encodes both form I and form II of rubisco. These unique features make *R. palustris* an ideal microorganism to be considered a biotechnological chassis for further metabolic engineering (7). However, the complex interplay among these exclusive features and other metabolic pathways of *R. palustris* is still not properly understood and a systems-level analysis is required to understand such interplay.

One widely accepted systems-level investigation tool is the genome-scale M model (19). Initial efforts of reconstructing M models of purple nonsulfur bacteria (PNSB) were limited to the specific metabolic pathways of interest, such as central carbon metabolism (20), and electron transport chain (21). However, those pathway-specific M models did not have a wider resolution to capture the overall metabolic landscape of PNSBs. To overcome that, comprehensive M models were reconstructed for PNSB strains, including *Rhodobacter sphaeroides* (22) and *R. palustris* (23). Recently we further refined the *R. palustris* M model by integrating the annotated metabolic pathways for *p*-coumarate and coniferyl alcohol and validated it by using the experimental data on gene essentiality and metabolic flux analysis for growth under different carbon sources (24). Although these M models were useful to study different metabolic features of PNSB, the inherent lack of quantitative characterization of macromolecular machinery synthesis (MMS) (25), also known as the ribosome, may lead to incorrect predictions of biological scenarios, such as inaccurate reaction flux and multiple equivalent cellular phenotypic states (26, 27). These inaccuracies can lead to an erroneous understanding of the overall metabolic and regulatory features of an organism and can negatively impact the design-build-test-learn cycle for metabolic engineering application.

One of the ways to overcome incorrect predictions of biological scenarios is to use resource allocation-based models, such as whole-cell modeling (28), resource balance analysis (29), and the metabolic and expression (ME) modeling approach (30). Recently the ME modeling approach was used to answer fundamental biological questions regarding the genotype-phenotype relationship. ME model is a resource allocation-based model that includes not only the stoichiometric metabolic reactions but also quantitative MMS information (31). As input, ME models require the conditions of a steady-state environment and can then output everything which can be predicted through an M model, such as predictions for maximum growth rate, substrate uptake, by-product secretion, and metabolic fluxes. In addition, the ME model can exclusively predict gene expression levels, protein levels, and nonlinear growth curves (31). A ME model utilizes a growth optimization function along with coupling constraints that tie flux to transcriptional and translational reactions in the model. These constraints are functions of the growth rate. By including these constraints, ME models set limitations on fluxes based on transcription as well as translation reactions. Thus far, ME models were developed only for a few organisms. These models were used to accurately predict the cellular composition and gene expression of *Thermotoga maritima* (32), fermentation profile of *Clostridium ljungdahlii* (33), overflow metabolism of *Saccharomyces cerevisiae* (34), and multiscale phenotype, enzyme abundance, and acid stress of *Escherichia coli* (35–37). An ME model for *R. palustris* can also be very useful in answering fundamental biological questions, such as growth profiling, isozyme

expression prediction, regulation of electron distribution between competing metabolic modules, and temperature regulation of different enzymes.

In this work, the first-ever ME model was reconstructed for *R. palustris*. The ME model was able to satisfactorily recapitulate the experimental transcriptomics and proteomics observations from the literature (38). Acetate, succinate, butyrate, and *p*-coumarate were then used as carbon sources to characterize the growth profile of *R. palustris*, which closely matched with experimental growth rate data. In addition, it predicted a diminishing rate of carbon fixation at the theoretical maximum growth rate and consequently predicted malate dehydrogenase and glycerol-3 phosphate dehydrogenase as alternate electron sinks. Furthermore, the ME model identified ferredoxin as a key regulator in distributing electrons between major redox balancing pathways, such as carbon and nitrogen fixation. Later, the modeling framework was able to capture experimentally observed temperature regulation of different nitrogenase enzymes by varying turnover rate of nitrogen fixation reactions. Overall, this modeling framework demonstrated a bottom-up systems-biology approach that can be used to predict and analyze the cellular physiology of *R. palustris*, thereby providing an opportunity to generate experimentally testable hypotheses.

## RESULTS AND DISCUSSION

**Metabolic and expression model development.** To reconstruct the ME model, our previously reconstructed M model of *R. palustris*, *iRpa940* (24), was used as a template. To reconstruct the ME model, gene-protein-reaction (GPR) relationships for all the reactions were manually curated from the complete genome sequence of *R. palustris* (2). Transcription and translation reactions were added for reactions for which GPR relationships were available. These transcription and translation reactions in the ME model did not quantify the amount of mRNA and proteins. However, these predict the rate of production for each of the transcription and translation reactions, which can be considered the proxy of the amount of mRNA and proteins. Reactions for which GPR associations are not available, it was assumed that an average bacterial enzyme with 31.09 kDa molecular weight (39) catalyzed each reaction. Overall, the ME model contains 1397 reactions, 1483 unique metabolites, and 826 genes. A comparison between these matrices between the M model and ME model can be found in Table S1. From the comparison, the M model could only predict the growth rate where more nutrient results in more growth, also known as the exponential growth phase. However, the ME model could predict the exponential growth phase like the M model. In addition, it could predict the stationary growth phase where the growth rate was limited by the nutrient uptake and resource allocation. Furthermore, the M model is in good agreement with the ME model for the exponential growth phase. A similar observation of growth rate prediction between the M model and the ME model was also made in the previous ME model of *E. coli* (35). Fig. 1 shows the workflow of the ME model reconstruction.

In *R. palustris*, there are two different rubisco enzymes, form I and form II. Form I rubisco ($L_8S_8$) is comprised of eight active large subunits ($L_8$) and active eight small subunits ($S_8$) (40) and is encoded by two genes, *rpa1559* and *rpa1560* (2). On the other hand, form II rubisco ($L_2$) is comprised of two active large subunits both encoded by *rpa4641* (2). Of the two forms of rubisco, form I has a higher molecular weight compared to form II (41, 42) and, therefore, requires more carbon investment to synthesize. Because rubisco is one of the most abundant enzymes in nature (43), the kinetics of this enzyme has been determined for multiple organisms (40, 44). For different rubisco enzymes, it was shown that although form I has higher molecular weight and more carbon investment cost, form II has a higher catalytic turnover rate ($k_{cat}$) per active site compared to form I (40). Evolutionary selection has played a major role in this counterintuitive observation (45, 46). Early in the earth's history, the concentration of carbon dioxide was higher in the atmosphere and as a result form, II rubisco evolved with a lower selectivity and higher $k_{cat}$ for carbon dioxide (40). With increasing amounts of oxygen in the earth's atmosphere, form I evolved with a much higher selectivity for carbon dioxide but with a lower $k_{cat}$ (40). Because $k_{cat}$ values for *R. palustris* are not available, to account for these evolutionary selections, the $k_{cat}$ values were set to 3.7 $s^{-1}$ active site$^{-1}$ (form I) and 6.6 $s^{-1}$ active site$^{-1}$ (form II) based on the

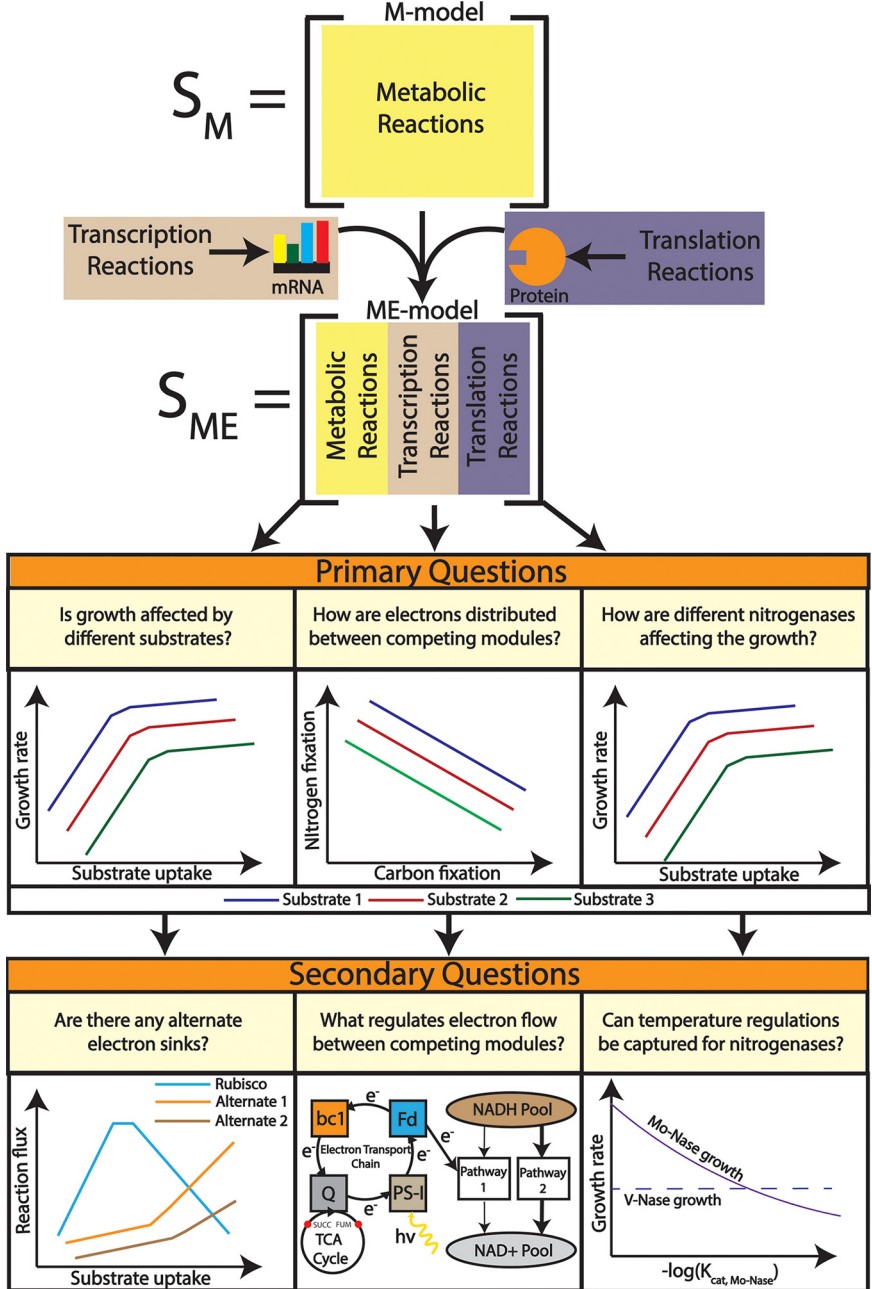

**FIG 1** Workflow followed to reconstruct the ME model from a previously published M model of *R. palustris*. Transcription and translation reactions were added on top of the metabolic reactions to come up with an ME modeling framework. The ME modeling framework was used to characterize growth rate profiling, competing for metabolic modules, and nitrogenase enzyme activity. From these characterizations, inferences regarding alternate redox balancing, ferredoxin regulation, and temperature regulation of nitrogenase enzymes were gathered.

measurements from other phylogenetically close (47) PNSB strains (*Rhodobacter capsulatus* (44) and *Rhodospirillum rubrum* (40), respectively).

*R. palustris* has three different nitrogenase enzymes, Mo-Nase, V-Nase, and Fe-Nase. Each enzyme is encoded by a series of genes (2) (Mo-Nase by *rpa4602* to *rpa4633*, V-Nase by *rpa1370* to *rpa1380*, and Fe-Nase by *rpa1435* to *rpa1439*). Unlike rubisco, $k_{cat}$ values of different nitrogenase are not available for *R. palustris* or any other PNSBs. Therefore, the calculated surface accessible surface area (SASA) of each nitrogenase enzyme was used to normalize the mean $k_{cat}$ value, as discussed elsewhere (35) (see

Materials and Methods). These normalized $k_{cat}$ values were used to define three independent nitrogen fixation reactions.

Both above-mentioned enzymes, nitrogenase and rubisco, play a pertinent role in maintaining the cellular redox balance during the photoheterotrophic growth of *R. palustris* by regenerating oxidized cofactors (48). When the ME model was used to simulate the photoheterotrophic growth of *R. palustris*, among three different nitrogenase enzymes, it predicted the expression of Mo-Nase only, which is consistent with the literature (49, 50). For the same photoheterotrophic growth conditions, between two different forms of rubisco enzymes, the model predicted only the expression of form II rubisco. Although expression of only rubisco form II was expected based on its lower carbon cost and higher efficiency, literature evidence suggested coexpression of both forms of rubisco during the photoheterotrophic growth of *R. palustris* (51). The same work suggested that rubisco form I is responsible for providing cellular carbon and dominates under carbon dioxide limiting conditions, whereas rubisco form II balances the intracellular redox potential under carbon and electron abundant conditions (51). In addition, it was also found that the expression of the cbb*cbb* operons (responsible for coding both forms of rubisco) during phototrophic growth is highly dependent on the cellular carbon dioxide level (51). To incorporate these findings, a constraint was added to the ME model to coexpress both forms of rubisco based on the total carbon dioxide produced by *R. palustris* during photoheterotrophic growth (see Materials and Methods).

**Model validation using experimental transcriptomics and proteomics data.** To validate the prediction accuracy of the model, experimental transcriptomics and proteomics data were used to qualitatively verify whether the model can predict the direction of these experimental fold changes in different conditions. A previous study, which characterized the anaerobic growth of *R. palustris* by comparing the transcriptomic and proteomic profiles of cultures grown in the presence of *p*-coumarate and succinate as the sole carbon source, was used for the validation study (38). The study tested fold change of 4810 genes for *p*-coumarate catabolism considering succinate catabolism as the baseline condition using Affymetrix GeneChip Operating Software Version 1.4 and Cyber-T program. The transcriptomic analysis, which is based on relative transcript abundance, resulted in 369 differentially expressed genes, among which 61 were metabolic genes. Similarly, the proteomic analysis resulted in 341 differentially expressed proteins, among which 67 can act as enzymes. In both transcriptomics and proteomics data sets, nonmetabolic genes/proteins have functions, such as signaling, chromosomal replication, and circadian rhythm (Table S2).

To generate both gene and protein expression information for the same two conditions of the above-mentioned study (38), the ME model was simulated for two points where total rubisco flux was maximal for the *p*-coumarate and succinate uptake, respectively. It was previously reported (48) that carbon fixation is required to maintain redox balance in *R. palustris*. Therefore, a higher growth rate is associated with higher reduced cofactor production, leading to higher rates of carbon fixation. As a result, the decreasing carbon fixation flux with increasing growth (Fig. 2) is a theoretical feature predicted by the ME model. All the experimentally observed and differentially expressed genes and proteins are available in the model. However, for reactions catalyzed by multiple isozymes, the ME model only predicted the most efficient isozyme based on the $k_{cat}$ and molecular weight. As a result, out of these experimentally observed 61 genes and 67 enzymes with metabolic functions, we could only compare 25 genes and 37 enzymes with the experimental data. To eliminate the possibility of alternate solution space, we used the minimization of the total sum of fluxes as an outer objective (52). Hence, the variability of each of the reactions was very tight and the fluxes of the reactions were distributed in a way such that the total amount of enzyme requirements are minimized. Similar tight variability of reaction fluxes was also noticed in the previously published ME model for *E. coli* (35).

As part of the transcriptomics data validation, out of 25 genes, the ME model was able to predict correct gene expression fold change for 23 genes. The model could not

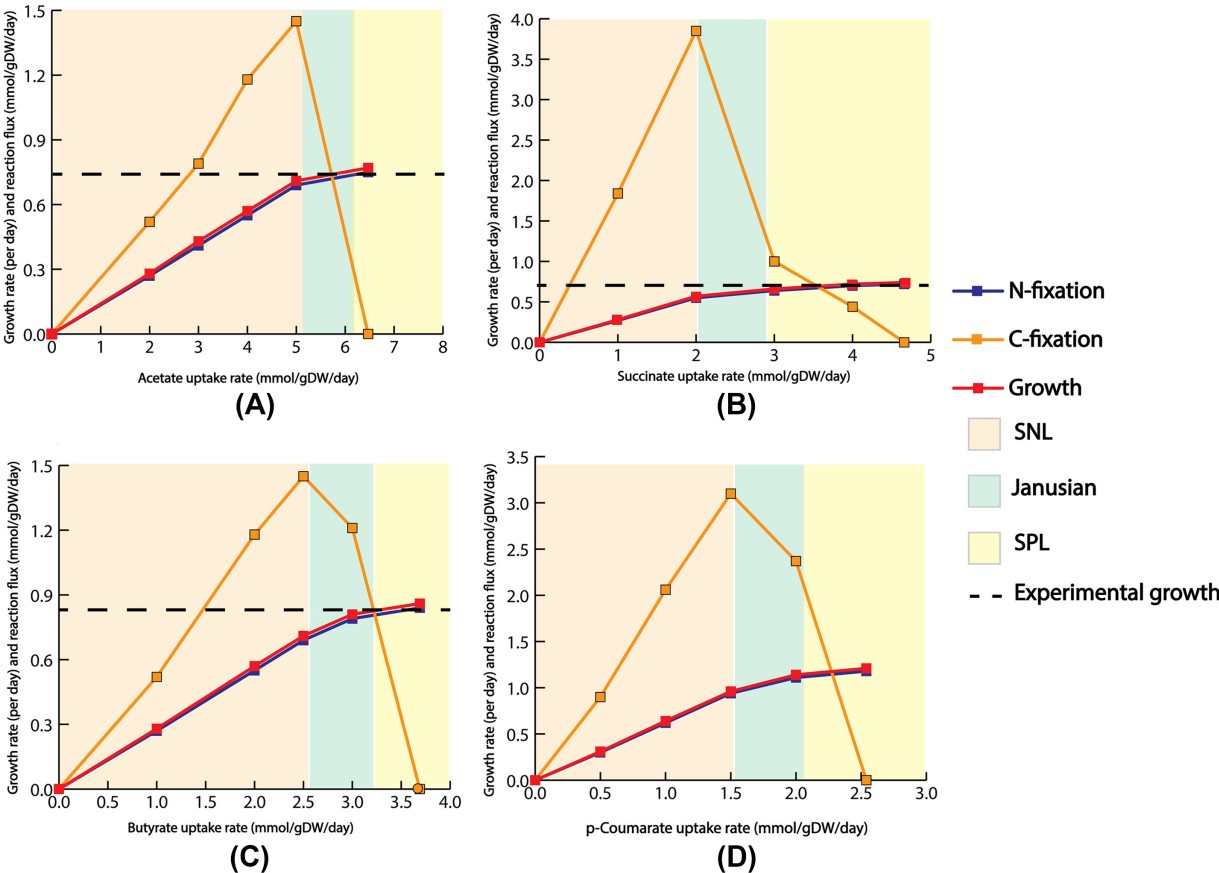

**FIG 2** Strictly nutrient-limited (SNL), Janusian, and strictly proteome-limited (SPL) regions for (A) acetate, (B) succinate, (C) butyrate, and (D) *p*-coumarate. The growth rate with respect to different substrate uptakes follows a nonlinear pattern. Flux through nitrogen fixation reaction also followed a similar pattern to the growth rate. Carbon fixation reached a peak in the Janusian region and then diminished in the theoretical maximal growth.

predict the downward fold change of 3-oxoacyl-acyl carrier protein reductase (*rpa3304*) and the 50S ribosomal protein (*rpa0918*). *rpa3304* is one of the genes to convert malonyl-CoA to biotin (53). Biotin is a part of the *R. palustris* cell membrane (54). In Fig. 2, it can be seen that p-coumarate supported a higher growth rate than succinate. Thus, the ME model predicted an upward fold change of *rpa3304* for *p*-coumarate catabolism compared to succinate catabolism. The composition of biotin in the cell membrane may be different in different conditions. However, in the ME model, only protein and nucleotide compositions change with different conditions, while those of cell wall components remain constant (30). This may have caused the mismatch. For incorrect fold change prediction of 50S ribosomal protein, missing reactions, the lack of regulatory mechanisms, and inaccurate $k_{cat}$ data may have played a role. Besides these factors, uncertainty arising from the different parameters in the ME model, such as the mass of rRNA per ribosome ($m_{rr}$), the molecular weight of average amino acid ($m_{aa}$), and the fraction of RNA that was rRNA ($f_{rRNA}$), which were used from the previously published model ME model of *E. coli* (35) and may have contributed to these incorrect predictions.

For proteomics data validation, out of 37 enzymes, the ME model was able to correctly predict the fold change for 25 enzymes. The ME model could not correctly predict the downward fold change of 12 different enzymes (Table S2). The number of incorrect predictions is higher in the validation of the proteomics data compared to the transcriptomics data. One of the inherent weaknesses of the ME model is unless additional constraints are imposed, the amount of protein is proportional to the amount of transcript, which may not be true in some instances (55). These incorrectly

**TABLE 1** Normalized growth rate for different substrate uptakes

| Substrate | Experimentally observed growth rate (day$^{-1}$) | The growth rate from the ME model (day$^{-1}$) | Substrate uptake for expt growth rate from model (mmol. gDW$^{-1}$ day$^{-1}$) |
|---|---|---|---|
| Succinate | 0.70 | 0.74 | 4.66 |
| Acetate | 0.74 | 0.77 | 6.47 |
| Butyrate | 0.82 | 0.86 | 3.69 |
| *p*-coumarate | –[a] | 1.21 | 2.54 |

[a]- indicates that no experimentally observed growth rate data is available for *p*-coumarate uptake.

predicted enzymes are mainly associated with purine and pyrimidine metabolism, fatty acid metabolism, and lipopolysaccharide metabolism. These pathways are closely associated with the *R. palustris* biomass growth. Because *p*-coumarate supports more growth than succinate, the ME model allocated more proteins for these pathways to sustain biomass growth. There may be unannotated alternate metabolic pathways with less enzyme investment for producing purine, pyrimidine, fatty acid, and lipopolysaccharide when *p*-coumarate is utilized as the carbon source, thus causing these discrepancies. Because the ME model maximizes the biomass growth rate, such incorrect prediction can be considered an inherent weakness of the ME model.

Overall, despite these incorrect fold change predictions, the ME model was able to satisfactorily recapitulate the subgroup of transcriptomics and proteomics observations which matched with the experimental data with 92% and 68% accuracy, respectively (see Materials and Methods for accuracy calculation). The details of experimental and model predictions are in Table S2.

**Growth rate versus substrate uptake and alternate redox balancing strategies.** Upon the validation with available gene expression and protein abundance data, the model was used to examine how growth, carbon fixation, and nitrogen fixation rates varied with different substrate uptake rates. The goal of this analysis was to investigate how reducing power entering the cell through organic carbon sources gets partitioned into biomass, carbon dioxide fixation, and nitrogen fixation. To perform the analysis, acetate, succinate, butyrate, and *p*-coumarate were used as the substrates. Previous studies have shown that photoheterotrophic growth of *R. palustris* on acetate, succinate, and butyrate is associated with increased cellular redox stress based on the oxidation state of different substrates (56). Hence, these substrates were chosen as they cover a wide range of oxidation states. Here, succinate (+0.5) and acetate (0) have higher oxidation states compared to *R. palustris'* biomass (−0.13) (49), whereas butyrate (−1) and *p*-coumarate (−0.22) have lower oxidation states (49). These oxidation states are based on neutral molecules.

In the ME model, the growth rate is a nonlinear function of substrate uptake rate and eventually reaches a theoretical maximum growth rate (Fig. 2). This behavior is consistent with known microbial empirical growth models, such as Monod growth kinetics (57) and microbial slow growth kinetics (58). Previous work has suggested three distinct growth regions as a function of substrate uptake rate. Strictly nutrient-limited (SNL), Janusian, and strictly proteome-limited (SPL) (35). Growth in the SNL region depends heavily on nutrient uptake and adding more nutrients results in more growth. In this region, the relationship between growth rate and substrate uptake is similar to the prediction made from M models. Contrary to the SNL region, growth in the SPL region (also known as the nutrient excess condition) is limited by the physiological constraint of protein production and catalysis. Janusian growth is the region where a transition from SNL to SPL takes place. A recent experimental study (49) had characterized the growth of wild-type (WT) *R. palustris* for acetate, succinate, and butyrate, respectively, under nitrogen-fixing conditions. Table 1 compares experimentally observed growth rates and those predicted by the model. The growth rate and order predicted by the ME model for succinate, acetate, and butyrate closely followed the experimental growth rate and order. Compared to other substrates, the ME model predicted a significantly higher growth rate on *p*-coumarate. One of our

previous works (7), which experimentally examined different strategies for PHB production under non-nitrogen fixing conditions, also showed significantly higher growth in p-coumarate compared to butyrate and acetate. It was previously reported (7) that, *p*-coumarate consumption led to more ATP production compared to acetate, succinate, and butyrate and thus was able to support more growth.

Theoretical growth rates predicted by the ME model were slightly higher than the experimental growth rates for all tested substrates (6% for succinate, 5% for butyrate, and 4% for acetate). The cell has many more layers of physiological regulations, such as transcriptional regulation, allosteric regulation, posttranscriptional regulations, and single nucleotide polymorphisms, which were not captured in the ME modeling framework. Overall, a growth rate comparison between the ME model prediction and experimental study reveals that similar to *E. coli* (35), optimum resource allocation dictates metabolic activities for *R. palustris*. Table S3 has the theoretical maximum growth rates for different amounts of substrate uptakes.

After characterizing the growth rate with different substrate uptakes, the ME model was used to characterize nitrogen and carbon fixation rates as a function of substrate uptake. For nitrogen fixation, the reaction's activity followed a similar trajectory as growth versus substrate uptake (Fig. 2). Different studies have shown that during photoheterotrophic growth, among three different nitrogenase (Mo-, V-, and Fe-Nase) isozymes encoded in *R. palustris'* genome, Mo-Nase is exclusively expressed (49, 50). *A. vinelandii*, which has three different nitrogenases, also exclusively expresses the Mo-Nase (59). The ME model predicted exclusive expression of Mo-Nase during growth on all four carbon sources. The expression of nitrogenase may be dictated by its ATP requirements because Mo-Nase required the least amount of ATP among the three nitrogenases. In addition, the temperature of the assay played a role in the expression of different nitrogenases as discussed later.

Next, carbon fixation was also characterized with respect to substrate uptake. Unlike nitrogenase, which closely followed the trajectory of the growth rate, carbon fixation reached a peak flux at the start of the Janusian region. In the SPL region, when growth is proteome limited, *R. palustris* optimized protein production to sustain the growing biomass demand. As the cell approached the theoretical maximal growth, more ribose-5 phosphate was needed to sustain the increasing demand for nucleotides and lipopolysaccharides. To meet that demand at the theoretical maximum growth, the ME model predicted that *R. palustris* decreased the expression of phosphoribulokinase (*rpa4645*) and redirected flux toward ribose-5 phosphate production (Fig. 3).

During photoheterotrophic growth under nitrogen-fixing conditions, carbon and nitrogen fixation played a major role in maintaining cellular redox balance. However, in the SPL region, because the reaction flux of carbon fixation diminished at the theoretical maximum growth, the ME model predicted two potential candidates to maintain cellular redox balance: malate dehydrogenase and glycerol-3 phosphate dehydrogenase, in addition to nitrogen-fixing reaction. Malate dehydrogenase uses NAD+/NADH as cofactors and was encoded by *rpa0192*. Similarly, glycerol-3 phosphate dehydrogenase used NAD+/NADH as cofactors and was encoded by *rpa4410*. During the switch from the SNL to the SPL region, at the point where carbon fixation started to diminish, both malate dehydrogenase and glycerol-3 phosphate dehydrogenase fluxes started to increase (Fig. 4). At the theoretical maximum growth, flux through malate dehydrogenase and glycerol-3 phosphate dehydrogenase reached its maximum. Malate dehydrogenase also plays a role in maintaining redox balance in several other Gram-negative bacteria, such as organisms, including *E. coli* (60), and *Corynebacterium glutamicum* (61). Glycerol-3 phosphate dehydrogenase was one of the key enzymes in fatty acid biosynthesis. It was suggested that other biosynthetic pathways, such as fatty acid biosynthesis, could offer flexibility contributing to the redox balance for photoheterotrophically grown *R. rubrum* (62). In addition, several other organisms, such as *S. cerevisiae* (63) and *Kluyveromyces lactis* (64), showed evidence of using glycerol-3 phosphate dehydrogenase to maintain redox balance.

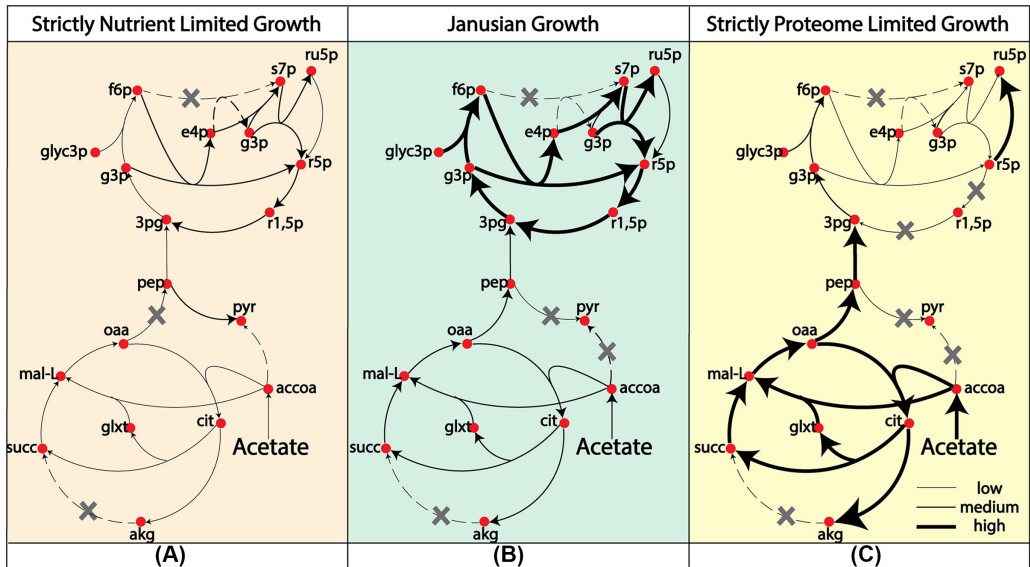

**FIG 3** Metabolic activities in the (A) strictly nutrient-limited (SNL) growth, (B) Janusian growth, and (C) strictly proteome limited (SPL) growth. In the theoretical maximum growth, at the SPL region, flux through carbon fixation diminished and reaction flux from ribulose-5 phosphate (cpd00171) to ribose-5 phosphate (cpd00101) significantly increased. The increased biomass growth demand can be met by the precursors from the TCA cycle, which showed a significant increase in reaction flux compared to Janusian growth and SNL. Gray crosses indicate zero reaction flux.

**Carbon fixation versus nitrogen fixation – competing metabolic modules for redox balance.** During photoheterotrophic growth, *R. palustris* performs cyclic photophosphorylation (2, 21), which meant that electrons from photosystem I (PSI) got transported through ferredoxin, and the $bc_1$ complex and recycled back to PSI through the oxidation and reduction of quinones (24) (Fig. 5). Because there were no terminal electron acceptors, this could cause an accumulation of reduced cofactors, resulting in impeded growth of the bacterium. To resolve this, *R. palustris* employed various electron sinks to maintain a cellular redox balance. Calvin–Benson–Bassham (CBB) cycle pathway (48), nitrogen fixation pathway (65), and PHB production pathway only under nitrogen starvation condition (7) were some of the prominent pathways. During photoheterotrophic growth, the redox-balancing mechanism consisted primarily of the CBB pathway (48) and nitrogen fixation pathway (65). The nitrogen fixation module became active when *R. palustris* was placed in a nitrogen-limiting environment. Experimental studies have suggested a link between carbon and nitrogen fixation that is intimately associated with the control of intracellular redox balance for different PNSBs, such as *R. palustris* (48), *R. capsulatus* (66), *R. sphaeroides* (65, 67), and *R. rubrum* (65). However, it is still not properly understood what factors decide the distribution of electrons in these two competing metabolic modules. Our previous study (24) indicated that with the increasing light uptake, the quinol oxidation rate also increases and the oxidation state of quinone acts as a feed-forward controller of the CBB cycle. Here, the ME model was used to further analyze the metabolic factors deciding the distribution of electron flux between carbon and nitrogen fixation in maintaining cellular redox balance.

To understand the electron distribution, a previous study eliminated rubisco activity in *R. palustris* by knocking out the relevant genes and found that the rate of nitrogen fixation did not vary significantly (48). Because CBB and nitrogen fixation pathways are two major redox balancing mechanisms, when rubisco was eliminated, the nitrogen fixation pathway was likely to carry an additional flux load to maintain cellular redox balance. Because this was not the case in the previous experimental study (48), it was suspected that there exists a metabolic bottleneck preventing additional reaction flux through the nitrogen fixation pathway. In the cyclic photophosphorylation of *R. palustris*, electrons get transported in a cyclical manner, and reduced ferredoxin supplies electrons to the nitrogen fixation pathway. Based on the availability of electrons, the nitrogen fixation pathway uses reduced cofactors to fix nitrogen. The more electrons supplied by ferredoxin, the more reduced cofactors will be used by nitrogen fixation

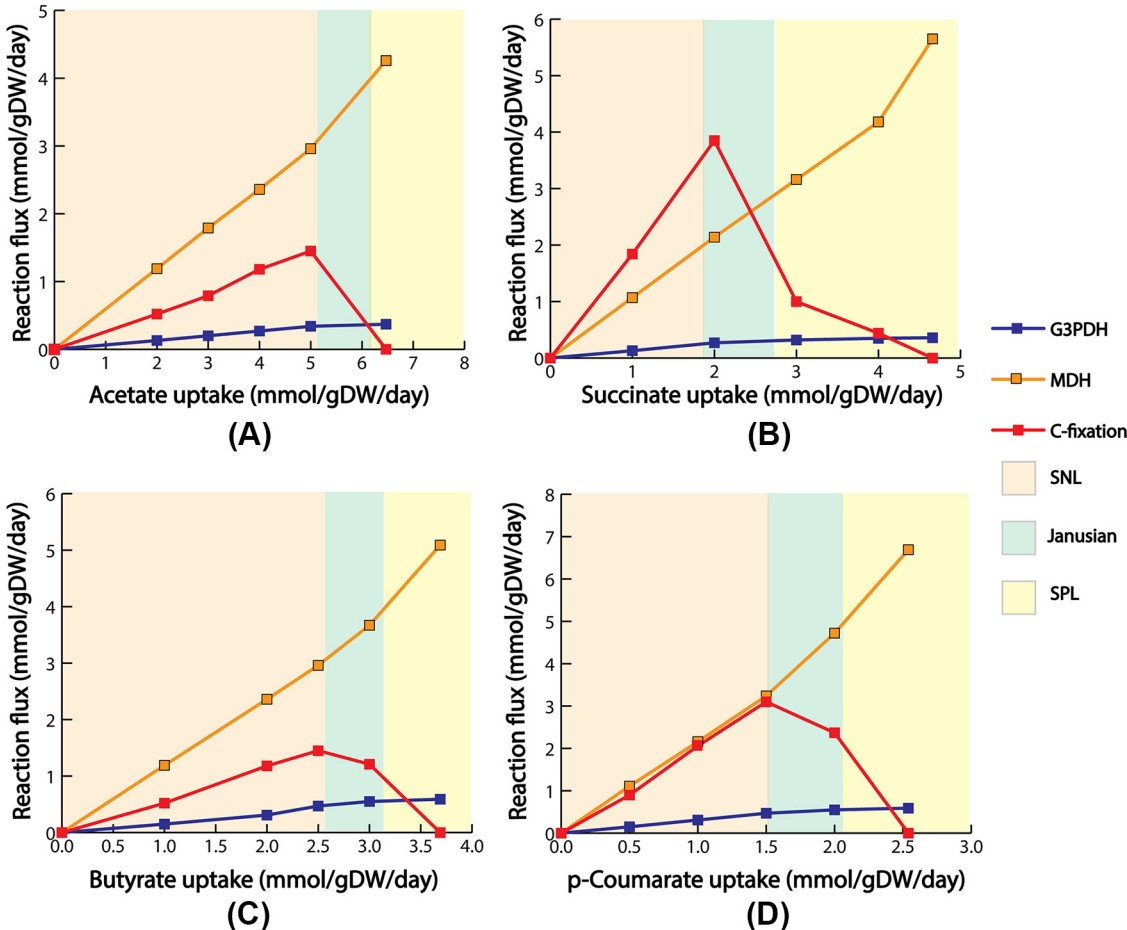

**FIG 4** Alternate electron sink for different substrates (A) acetate, (B) succinate, (C) butyrate, and *p*-coumarate (D). In the Janusian regions, flux through carbon fixation reaction started to diminish. With the diminishing carbon fixation flux, the ME model predicted two alternate electrons, malate dehydrogenase, and glycerol-3 phosphate dehydrogenase. Reaction flux through these alternate electron sinks reached its peak when flux through carbon fixation completely diminished at the theoretical maximum growth.

pathways. Hence, less reduced cofactors were available for the carbon fixation pathway to use. Hence, electron transport through ferredoxin (ETFD) could be a potential candidate for the previously discussed bottleneck. Another element that transports electrons to the nitrogen fixation pathway is flavodoxin. A previous study (68) indicated that flavodoxin is a prominent electron donor only when *R. palustris* is under iron starvation. Flavodoxin is an isofunctional flavoprotein present in *R. palustris*, which is induced and replaces ferredoxin under stress conditions. In this study, we did not study the growth of *R. palustris* under any starvation situation. Hence, it is expected the role of flavodoxin in the redox balance will be minimum.

To explore if ETFD was indeed the hypothesized bottleneck, the biomass growth, and substrate uptake rate were kept constant, and only flux through carbon fixation reaction was varied for increasing flux of electron transport through the ferredoxin reaction (ETFD). At first, flux through ETFD was fixed to the solution found by the ME model (indicated by the red line in Fig. 6). The flux through the nitrogen-fixing reaction remained constant with changing flux through the carbon fixation reaction. This finding confirmed the presence of the previously hypothesized bottleneck. Increasing flux through ETFD had varying effects on the rate of nitrogen fixation depending on the utilized carbon substrate. When the reaction flux through ETFD was set to values higher than the ME model solution (indicated by the yellow and blue lines in Fig. 6), a very small change in flux through nitrogenase was noticed for the growth of acetate. For the other carbon sources, when the reaction flux through ETFD was set to values

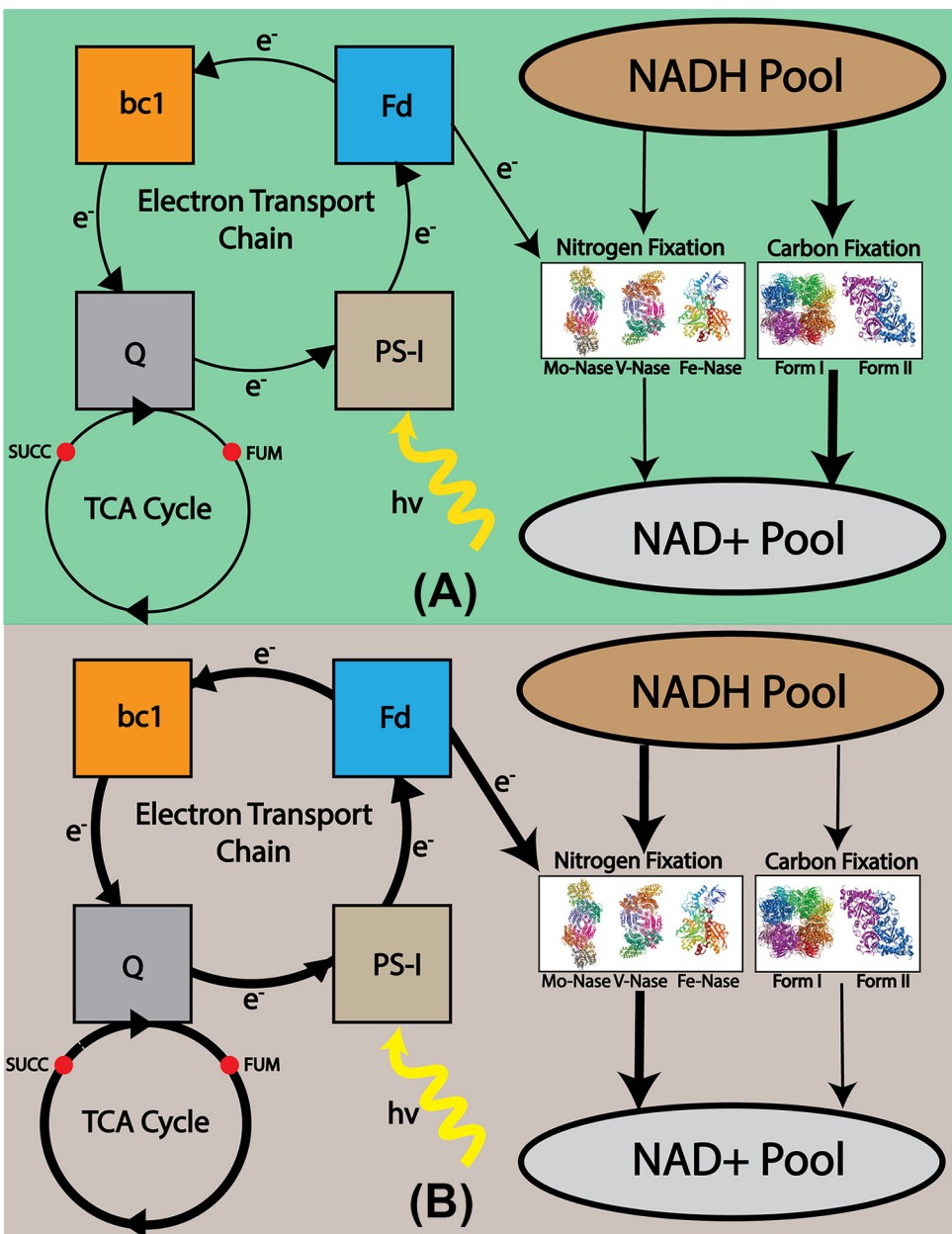

**FIG 5** Relation between cyclic photophosphorylation and electron distribution between carbon and nitrogen fixation. (A) Less electron through ferredoxin indicates less flux through nitrogen fixation and more flux through the carbon fixation pathway. As a result, NADH will be more oxidized through a carbon fixation reaction. (B) More electron through ferredoxin indicates more flux through nitrogen fixation and less flux through the carbon fixation pathway. As a result, NADH would be more oxidized through a nitrogen fixation reaction.

higher than the ME model solution (indicated by the yellow and blue lines in Fig. 6), a negative correlation was observed between the carbon and nitrogen fixation reaction flux. When the metabolite pool size (see Text S1 for metabolite pool size calculation) was calculated for different cofactors, acetate produced less reduced cofactors per unit of substrate uptake compared to other substrates. As in this case, the fixed nitrogen is the sole source of nitrogen, cell prioritizes electron transport to nitrogenase rather than rubisco, whose primary function was to maintain the redox balance in the cell. Thus, the relation between carbon and nitrogen fixation was less visible for acetate. However, for succinate, butyrate, and *p*-coumarate, more reduced cofactors are produced per unit of substrate uptake. Thus, more electrons are available for the carbon

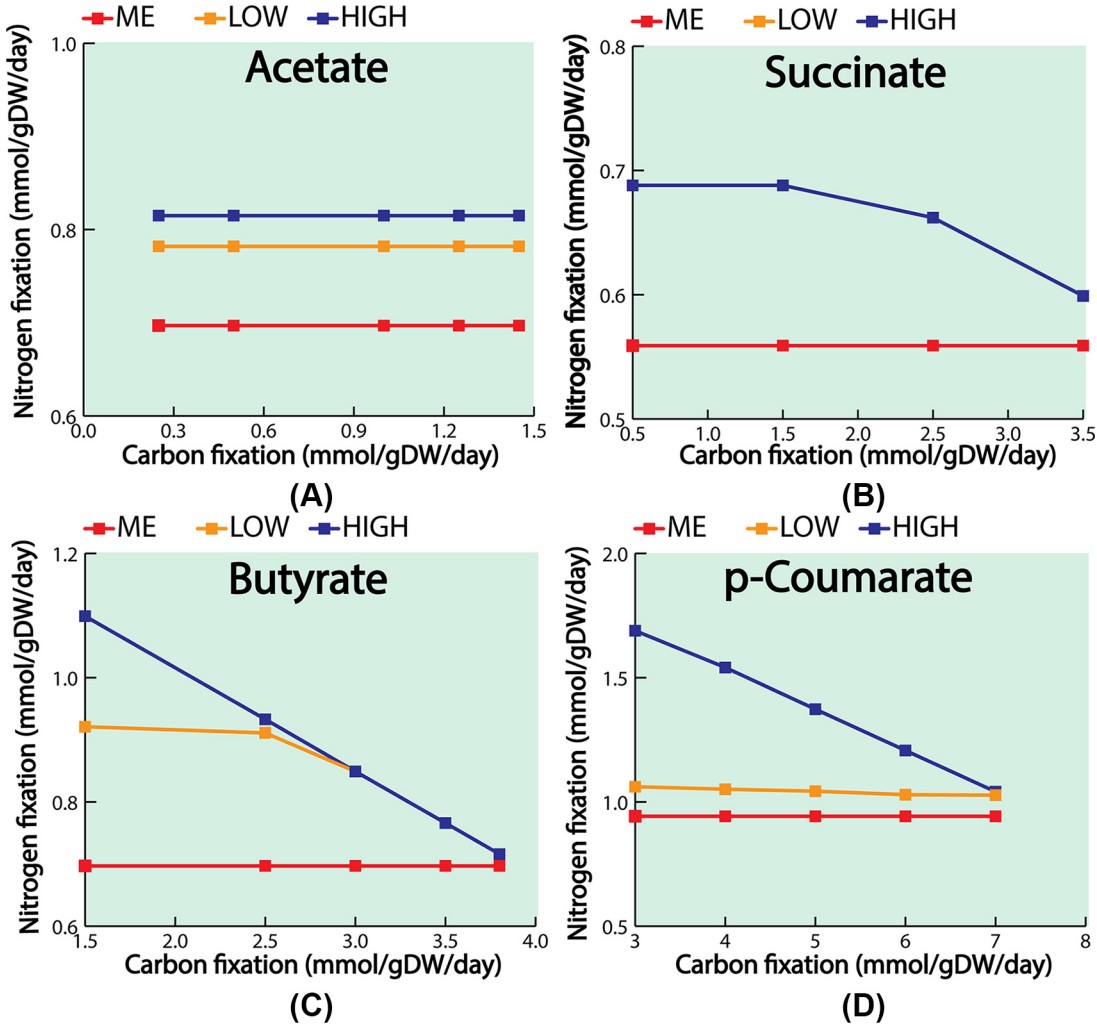

**FIG 6** Relation between carbon fixation and nitrogen fixation with different fluxes via electron transport through ferredoxin (ETFD) for different substrates: (A) acetate, (B) succinate, (C) butyrate, and (D) *p*-coumarate. Red color lines indicate the relation between carbon fixation and nitrogen fixation when flux through ETFD was set to the solution found in the ME model. Blue color lines indicate the relation between carbon fixation and nitrogen fixation when ETFD flux was set to a high value. Yellow color lines indicate the relation between carbon fixation and nitrogen fixation when the ETFD flux values were set between ME and high.

fixation pathway and the regulation is more visible when ETFD flux is higher for these substrates. These results indicated that reaction flux through ETFD may play a regulatory role in distributing electron flux between carbon and nitrogen fixation.

Similar regulation in electron transport between competing metabolic modules, such as respiratory pathways and electron transport, can be observed in model bacteria *E. coli* (69). A highly organized network of overlapping transcriptional regulatory elements regulated the flow of electrons by controlling the expression of different genes in *E. coli*, including genes involved in substrates uptake, control of mixed-acid fermentation pathways, and controlling cofactor biosynthesis. Further experimentation is required to establish a similar molecular level mechanism for ETFD regulation of electron distribution in competing pathways of *R. palustris*. The ETFD regulation, hypothesized in this study, could have profound implications for future metabolic engineering efforts of *R. palustris*. Specifically, this regulation could be exploited to increase hydrogen production from *R. palustris* to achieve energy sustainability goals.

**Characterization of Mo-, V-, and Fe-Nase nitrogenase enzymes.** Because ETFD was postulated to play a regulatory role in distributing electrons to the nitrogen-fixing pathway, the ME model was next used to characterize how these electrons were used

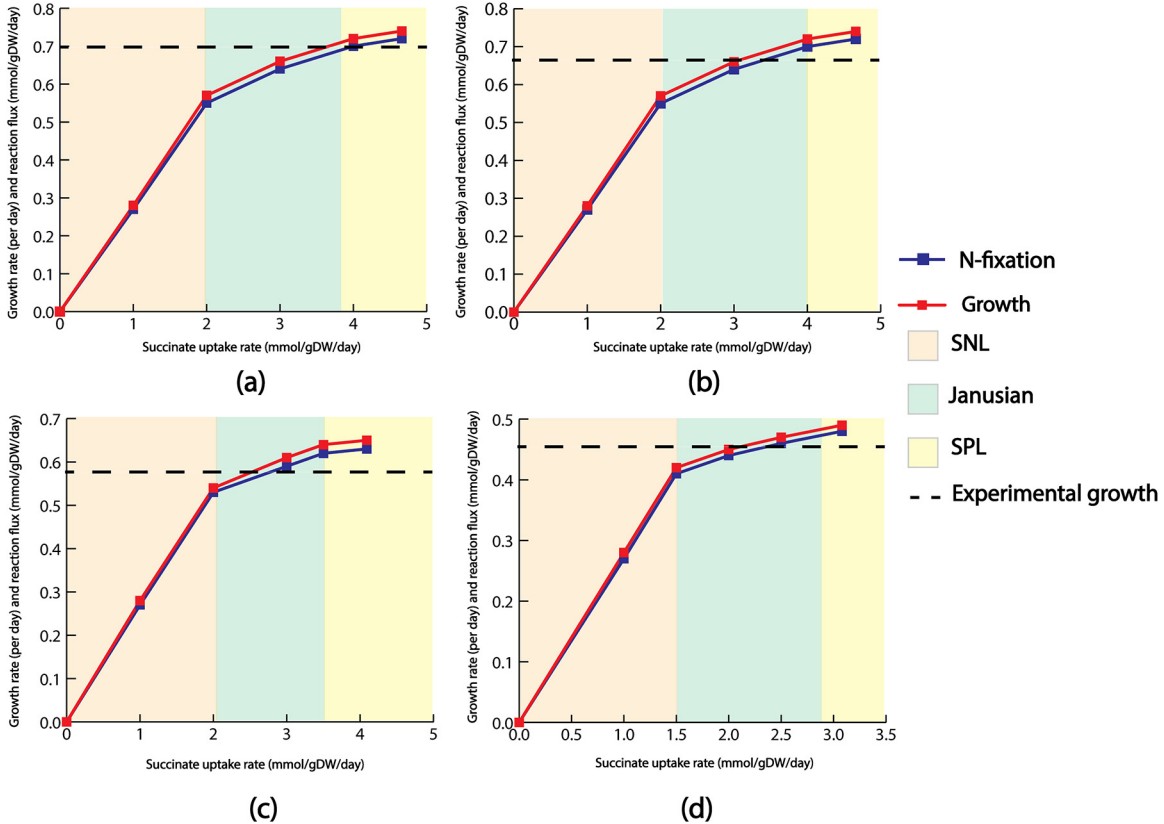

**FIG 7** Growth rate and nitrogen fixation rate for (A) WT *R. palustris*, (B) Mo-only mutant, (C) V-only mutant, and (D) Fe-only mutant for succinate uptake. For each case, growth rate and nitrogen fixation closely followed each other. The dotted line in each graph indicates the experimentally observed growth.

by different nitrogenase enzymes. First, growth was simulated for the WT *R. palustris* with succinate as the substrate. In this case, only Mo-Nase was expressed and the growth versus substrate uptake curve (Fig. 7A) followed the pattern identified in the literature (35). Exclusively expressing the Mo-Nase in the WT was also consistent with previous literature findings (49, 50). Next, the growth versus substrate uptake graphs (Fig. 7B to D) was developed for three different mutants of *R. palustris*, each expressing a single nitrogenase isozyme. When the theoretical maximum growths for these mutants were compared with WT, it was found that WT and the Mo-only mutant had the highest growth rate followed by the growth rate of V-only and Fe-only mutants. Compared with the experimental growth rate data from the literature (49) for WT, Mo-only, V-only, and Fe-only growths followed a similar pattern as predicted by the ME model (Fig. S1). Theoretically, the growth of the WT, V-only, and Fe-only mutant strains of *R. palustris* can be coupled with the ATP requirement because Mo-nase required the least and Fe-nase required the most amount of ATP for nitrogen fixation.

Contrary to the pattern observed for succinate uptake, when other carbon sources (acetate, butyrate, and *p*-coumarate) were used as the substrates, V-Nase exhibited higher growth compared to Mo-Nase, Fe-Nase, and even WT (Fig. S2 to S4). To explain this mismatch, a previous study observed that the Mo-Nase was less expressed at a lower temperature compared to the other isozymes, such as V-Nase (49). Fig. 8A qualitatively summarized this idea. Because the experimental values used in that study were generated at 19°C, it is possible that Mo-Nase has less selectivity toward fixing nitrogen rate than other substrates, such as nitragin ($N_2H_2$), methane, ethane, etc. The effect of decreasing assay temperature on the activity of nitrogenase is complex. It was reported (70) that for the Mo-Nase of *A. vinelandii*, the rate of nitrogen reduction at 10°C is very low despite continued hydrolysis of ATP. In the

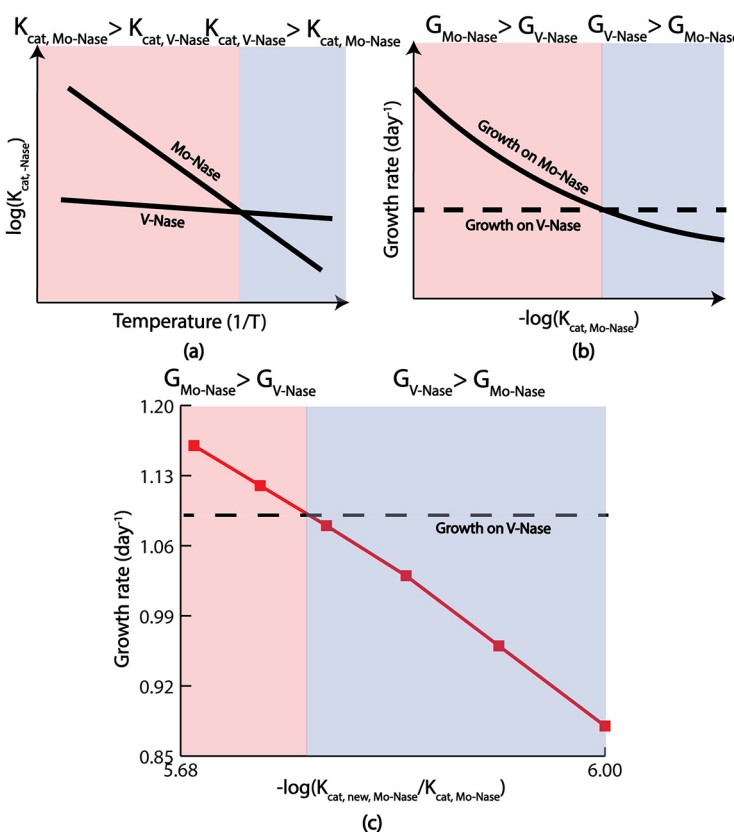

**FIG 8** A representation of temperature regulation of Mo-Nase and V-Nase, and the effect of $k_{cat}$ on the growth of Mo-only and V-only mutant. (A) From the literature, it was known that V-Nase had less sensitivity with respect to temperature compared to the Mo-Nase. The prediction from this ME model corroborated that finding. (B) Because $k_{cat}$ is a parameter that is a function of temperature, from the Arrhenius equation, we knew that by reducing the temperature, $k_{cat}$ was also reduced. With reducing $k_{cat}$ at one stage, growth for the Mo-only mutant fell below the growth of the V-only mutant, capturing the experimentally observed temperature regulation of Mo- and V-Nase. (C) A quantitative representation of the idea mentioned in (B).

case of Mo-Nase of *Klebsiella pneumoniae*, decreasing the temperature not only curtailed electron flux but also resulted in the preferential loss of activity toward nitrogen as a substrate compared with $H^+$ or ethyne ($C_2H_2$) (71).

The ME model was further used to investigate the decreased growth rate of Mo-Nase at a lower temperature. From the Arrhenius equation (72), it is known that the turnover rate of an enzyme, $k_{cat}$, increases exponentially with the increasing temperature. Because $k_{cat}$ is one of the temperature-sensitive parameters in this study, $k_{cat}$ values of Mo- and V-Nase were varied to see at what point the V-Nase growth rate exceeded that of Mo-Nase and Mo-Nase. At first, the $k_{cat}$ of V-Nase was increased to a very high value, but the growth rate of *R. palustris* fixing nitrogen through V-Nase was still lower than the WT and Mo-Nase. This tuning of $k_{cat}$ indicated that the sensitivity of V-Nase activity with respect to temperature was low. This finding is consistent with previously published experimental work on another Gram-negative bacteria, *Azotobacter chroococcum* (73). Later, the $k_{cat}$ of Mo-Nase was decreased to a very low value, and at that low $k_{cat}$, the growth rate of *R. palustris* fixing nitrogen through Mo-Nase was lower than the V-Nase and higher than the Fe-Nase, which is similar to the finding from literature (49). Therefore, by tuning the $k_{cat}$, the ME model was able to capture the experimentally observed temperature sensitivity of different nitrogenase enzymes. Fig. 8B qualitatively and Fig. 8C quantitatively summarized the effect of $k_{cat}$ on the growth of Mo-only and V-only strains of *R. palustris*.

**Conclusions.** In this work, an ME model of *R. palustris* was developed. Growth rates predicted by the ME model for different substrates were closely matched

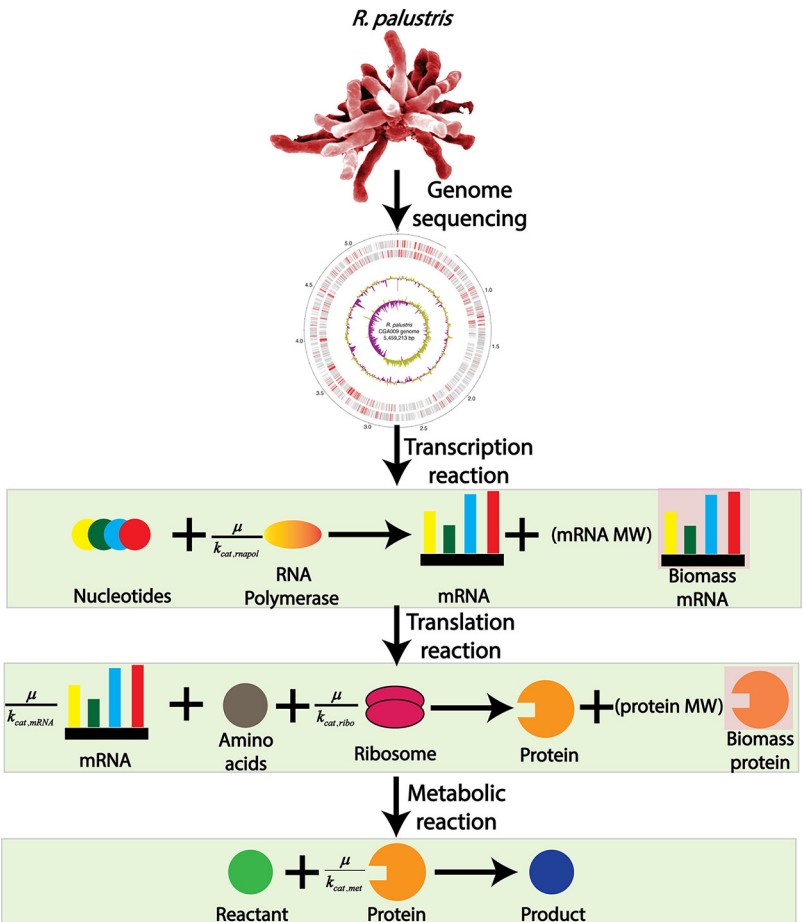

**FIG 9** *R. palustris* ME model reconstruction. In the M model, only metabolic reactions were incorporated to perform genome-scale metabolic modeling. However, in the ME modeling framework, transcription and translation processes were also incorporated, adding two separate layers of regulation for metabolic reaction. Each layer of regulation was coupled with the biomass growth through catalytic turnover rate and biomass growth. This process is known as coupling and the coupling parameter is in the form of $\mu/k_{cat}$, where $\mu$ indicates the growth rate and $k_{cat}$ indicates the catalytic turnover rate for that.

with experimental growth rate data. The ME model also predicted a diminishing carbon fixation at the theoretical maximum growth and subsequently malate dehydrogenase and glycerol-3 phosphate dehydrogenase as alternate electron sinks. Furthermore, the ME model postulated electron transport through ferredoxin as a key regulatory feature to distribute reduced cofactor pools between carbon and nitrogen fixation pathways. Finally, the ME modeling framework successfully captured experimentally observed temperature regulation of different nitrogenase enzymes.

Going forward, this ME model could be used as a powerful platform to further characterize different features of *R. palustris* metabolism. Specially characterizing a complete profile of environment-specific isozyme expressions and optimal protein allocation. Furthermore, this ME model can be used to design and fine-tune mutants of *R. palustris* for metabolic engineering purposes. One such application would be to produce PHB, a bioplastic precursor, which has the potential to replace petroleum-based plastics. Under anaerobic-photoheterotrophic growth of *R. palustris*, PHB can work as an electron sink (7). Our previous effort (7) successfully established three design strategies to select the ideal lignin breakdown products (LBPs) for commercial PHB production from *R. palustris*. This ME modeling framework could be further used to gain similar regulatory insights, as discussed here, on how electrons are distributed in PHB-producing pathways when different LBPs are used as the substrates.

## MATERIALS AND METHODS

**ME model of *R. palustris*.** In addition to the metabolic reactions from the M model, the ME model consisted of translation and transcription reactions (Fig. 9). To model transcription and translation reactions, a GPR association of each reaction was required. The initial GPR association was collected from the literature (24). Later, the GPR association was manually curated using the detailed genome annotation from literature (2).

This GPR association is in Table S4. The overall ME model reconstruction procedure was conducted in accordance with the COBRAme protocol (30), which is summarized in Fig. 2. The ME model is a multi-scale model (30). Hence, it required the addition of coupling constraints to relate different cellular processes to each other. The coupling constraints are in the form of $\mu/k_{cat}$. Here, $\mu$ was the growth rate and $k_{cat}$ approximated the effective turnover rate for the different macromolecules. Detailed mathematical descriptions for the $\mu/k_{cat}$ of different macromolecular processes and values of different parameters are in Text S1 and the original COBRAme protocol (30).

In terms of including tRNA charging reactions in the ME model (74), initiation was the rate-limiting step in the translation reaction rather than the elongation process. A similar result was obtained for the other prokaryotic and eukaryotic organisms (75, 76). Hence, we did not include the tRNA charging reactions in the ME model because it was not the rate-limiting step in the translation reaction. For protein translocation, the *R. palustris* model had only one compartment, cytosol, apart from the extracellular space. Hence, the need for translocation reactions was absent. Because DNA replication only ensures that the chromosome was copied at each cell doubling, the rate of this "reaction" would only be dependent on the growth rate and not on the growth condition. For posttranslational modifications, there is evidence of posttranslational modification for nitrogenase enzymes (77). But for most other proteins, posttranslational modifications were not known. If we had added posttranslational modifications only for nitrogenases, that would have introduced bias against nitrogenase enzymes compared to other enzymes from a resource allocation perspective. Hence, we chose not to add that bias to the model.

To calculate $k_{cat}$ for different enzymes, a mean $k_{cat}$ value of 234,000 day$^{-1}$ was used, which was reported for the photosynthetic cyanobacteria (78). This $k_{cat}$ was further modified for each enzyme based on the solvent-accessible surface area (SASA), following the same ME modeling framework (35). SASA was defined as the surface area of an enzyme that was accessible to a solvent. In addition, a previous study (79) reported a correlation between SASA and the molecular weight of the enzyme as follows: SASA = (molecular weight of the enzyme)$^{3/4}$. Overall, the following equation was used to calculate $k_{cat,enzyme}$ for each enzyme, based on the mean turnover rate ($k_{cat,\ mean}$), mean SASA ($SASA_{mean}$), and SASA for the specific enzyme ($SASA_{enzyme}$): $k_{cat,enzyme} = k_{cat,\ mean} \times SASA_{enzyme}/SASA_{mean}$. For transcription reactions, RNA polymerase was needed to produce the required mRNA for protein production. RNA polymerase of *R. palustris* consisted of five subunits: two alpha ($\alpha$) subunits, a beta ($\beta$) subunit, a beta prime subunit ($\beta'$), and a small omega ($\omega$) subunit (80). In the model, each subunit was synthesized to form the RNA polymerase. Later, these RNA polymerases transformed different nucleotides into mRNA.

For translation reactions, rRNA was required to transform amino acids into different proteins. *R. palustris* utilized 70S ribosomes, each consisting of a small (30S) and a large (50S) subunit (81). The large subunit was composed of a 5S RNA subunit (120 nucleotides), a 23S RNA subunit (2900 nucleotides), and 31 proteins. The small subunit was composed of a 16S RNA subunit (1542 nucleotides) and 21 proteins (81).

For each transcription or translation reaction in the ME model, an amount of a biomass protein and biomass mRNA were produced with a stoichiometry equal to the molecular weight (in kDa) of the protein or mRNA being made. Fig. 9 shows an example of this where the translation reaction produces both the catalytic protein and the biomass protein. Similarly, the transcription reaction produced mRNA required for the protein synthesis and biomass mRNA. The biomass protein and mRNA participated in the ME model biomass dilution reaction, restricting the total biomass components production equal to the rate of biomass dilution.

Transcription and translation reactions were included for all reactions for which GPR was available. For the remaining pathways, an enzyme was used with an average length of 283 amino acids and a molecular weight of 31.09 kDa based on the literature (39).

To capture the differential expression of the carbon fixing isozymes, a constraint was added to the ME model to account for the coexpression of both rubisco form I and form II as follows:

$$v_{rubisco\,I} = \left[ -\sum v_{CO_2} + \frac{\mu}{\mu_{max}} \sum v_{CO_{2,max}} \right] \times \frac{k_{cat,\,rubisco\,I}}{k_{cat,\,rubisco\,II}} \tag{1}$$

In equation (1), $v_{rubisco\ I}$ represent the expression of rubisco form I, which is a function of carbon dioxide generation ($\sum v_{CO_2}$), growth rate ($\mu$), theoretical maximum growth rate ($\mu_{max}$), carbon dioxide generation at the theoretical maximum growth rate ($\sum v_{CO_{2,max}}$), and effective catalytic rate of rubisco form I ($k_{cat,\ rubisco\ I}$) and rubisco form II ($k_{cat,\ rubisco\ II}$).

The main source of ATP production in *R. palustris* is photosynthesis (24). Hence, for each the substrate, the maximum ATP production by the ME model was capped by its photosynthetic yield according to the following equation proposed in the literature (7):

$$v_{PSII}^{S} = v_{PSII}^{ace} \frac{\varnothing_{PSII}^{S}}{\varnothing_{PSII}^{ace}} \tag{2}$$

Here, "S" and "ace" refer to different substrates and acetate, respectively. Also, $\varnothing_{PSII}^{S}$ and $\varnothing_{PSII}^{ace}$ refer to the photosynthetic yield of different substrates and acetate, respectively. Photosynthetic yields for different substrates are calculated from literature (7) and provided in Table 2 below.

**TABLE 2** Maximum ATP production rate calculated from photosynthetic yield for substrates

| Substrates | ATP production rate (mmol. gDW$^{-1}$ day$^{-1}$) |
|---|---|
| Acetate | 54.0 |
| Succinate | 45.7 |
| Butyrate | 56.7 |
| *p*-coumarate | 85.4 |

**Accuracy calculation in the validation study.** In the validation study, using the ME model, aerobic growth of *R. palustris* was simulated with *p*-coumarate and succinate as sources of carbon and $(NH_4)_2SO_4$ as a sole source of nitrogen. From the ME model, fluxes of transcriptomics and proteomics reactions were calculated for both carbon sources. Considering the transcriptomics and proteomics reaction fluxes for succinate uptake as the baseline condition, fold changes for all the gene expressions and protein were calculated for *p*-coumarate uptake. If the fold change is greater than 1, it was noted as upregulated. If the fold change is less than 1, it was noted as downregulated. Once the upregulated/downregulated fold changes of transcription and translation reactions were calculated and fold changes were compared with the literature (38). If both fold changes, from the ME model and the experimental study, showed the same direction (upregulated or downregulated) of fold change, then the prediction is correct. Otherwise, the prediction is incorrect. Accuracy was then calculated as a percentage between correct prediction and total predictions.

**Simulation tools and software.** Due to the large variability of scaling of different coefficients, the ME model was solved using the General Algebraic Modeling System (GAMS) version 24.7.4 with IBM CPLEX solver using the pFBA (52) algorithm. The GAMS files were run on a Linux-based high-performance cluster computing system at the University of Nebraska-Lincoln. For community use, an interface between GAMS Studio and NEOS Server is available to run the ME model without having to buy any license. Moreover, a python implementation is also available for interested users. Note that, python users will at least need to run the GAMS model once to generate the necessary input files. Text S2 has further instructions on how to run the GAMS and Python files using GAMS Studio and NEOS Server, without having to buy any license.

Because the ME model contains coupling coefficients. Hence, it is a nonlinear optimization problem. Thus, to find the optimal growth rate from the ME model, we parameterized the optimal biomass growth rate. To solve the optimization problem, we first set the biomass growth rate to a fixed value to make the problem linear. For a given substrates (acetate, succinate, butyrate, and *p*-coumarate) uptake rate, different biomass growth rates were simulated until the optimization problem became infeasible. Then the biomass growth rate just before the optimization problem become infeasible was reported as the optimal biomass growth rate.

**Data availability.** All code used in this work can be found in the following GitHub directory: https://github.com/ssbio/palustris_ME_model.

## SUPPLEMENTAL MATERIAL

Supplemental material is available online only.

**SUPPLEMENTAL FILE 1**, PDF file, 0.2 MB.
**SUPPLEMENTAL FILE 2**, XLSX file, 0.04 MB.
**SUPPLEMENTAL FILE 3**, XLSX file, 0.1 MB.
**SUPPLEMENTAL FILE 4**, XLSX file, 0.1 MB.
**SUPPLEMENTAL FILE 5**, XLSX file, 1.4 MB.
**SUPPLEMENTAL FILE 6**, PDF file, 0.1 MB.
**SUPPLEMENTAL FILE 7**, PDF file, 0.3 MB.

## ACKNOWLEDGMENTS

We gratefully acknowledge funding support from NSF Career Award grant number 1943310.

We declare no conflict of interest.

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
