## [Reviewer comments · Microbiology Spectrum]

Microbiology Spectrum

Characterizing the interplay of rubisco and nitrogenase enzymes in anaerobic-photoheterotrophically grown *Rhodospseudomonas palustris* CGA009 through a genome-scale metabolic and expression model

Niaz Chowdhury, Adill Alsiyabi, and Rajib Saha

Corresponding Author(s): Rajib Saha, University of Nebraska-Lincoln

Review Timeline:

Submission Date:

May 9, 2022

Accepted:

May 31, 2022

Editor: Jeffrey Gralnick

Reviewer(s): The reviewers have opted to remain anonymous.

Transaction Report:

DOI: <https://doi.org/10.1128/spectrum.01463-22>

May 31, 2022

Dr. Rajib Saha
University of Nebraska-Lincoln
Chemical & Biomolecular Engineering
1600 Vine Street
Lincoln, NE 68508

Re: Spectrum01463-22 (Characterizing the interplay of rubisco and nitrogenase enzymes in anaerobic-photoheterotrophically grown *Rhodospseudomonas palustris* CGA009 through a genome-scale metabolic and expression model)

Dear Dr. Rajib Saha:

Based on your responses and revisions to the prior round of review, your manuscript has been accepted, and I am forwarding it to the ASM Journals Department for publication. You will be notified when your proofs are ready to be viewed.

Sincerely,

Jeffrey Gralnick
Editor, Microbiology Spectrum

Journals Department
Text S1: Accept
Table S3: Accept
Table S4: Accept
FIG S1-S4: Accept
Table S1: Accept
Text S2: Accept
Table S2: Accept